# Novel Clinical Applications of 3D-Printed Highly Porous Titanium for Off-the-Shelf Cementless Joint Replacement Prostheses

**DOI:** 10.3390/biomimetics10090634

**Published:** 2025-09-20

**Authors:** Domenico Tigani, Luigigiuseppe Lamattina, Nicole Puteo, Cesare Donadono, Lorenzo Banci, Marta Colombo, Alex Pizzo, Andrea Assenza

**Affiliations:** 1Department of Orthopaedic Surgery, Ospedale Maggiore Carlo Alberto Pizzardi, 40133 Bologna, Italy; 2Medical Scientific Affairs, Permedica Orthopaedics, 23807 Merate, Italy; lorenzo.banci@permedica.it (L.B.); marta.colombo@permedica.it (M.C.)

**Keywords:** joint arthroplasty, titanium alloy, 3D printing, additive manufacturing, orthopaedic applications, highly porous lattice, cementless fixation, osseointegration, titanium–niobium nitride

## Abstract

In total joint replacement, early aseptic loosening of implants caused by inadequate initial fixation and late aseptic loosening due to stress shielding-related periprosthetic bone remodeling are the main causes of failure. Over the last two decades, additive manufacturing has been revolutionizing the design of cementless orthopaedic implants by enabling biomimetic, highly porous titanium structures that enhance bone ingrowth and osseointegration while reducing stress shielding. The synergy between optimized selective laser-melted highly porous titanium bearing components, ceramic-coated titanium articular surfaces, and vitamin E-stabilized polyethylene liners delivers several benefits essential for implant longevity: reliable initial fixation, improved biological fixation, reduced bone resorption caused by stress shielding, and lower osteolytic reactivity. These benefits have encouraged the synergetic use of these technologies in joint replacement in novel clinical applications. In recent years, novel off-the-shelf, 3D-printed, highly porous titanium implants have been introduced into hip and knee arthroplasty. These newly introduced implants appear to offer an innovative and promising solution, and are particularly indicated for young active patients, elderly patients with osteoporotic bones, and in complex cases. Future clinical research should confirm these novel implants’ superior results in comparison to the current state of the art in cementless joint replacement. The possibility of extending these technologies in the future to other clinical applications such as partial knee prosthesis is discussed.

## 1. Introduction

Total hip arthroplasty (THA) and total knee arthroplasty (TKA) are well-established treatments to restore mobility and reduce pain in patients with severe degenerative joint diseases. Despite continued improvements in prosthetic materials and designs, long-term revision rates remain significant [1]. Implant survival, which is expected to be lower in younger patients, highlights the critical importance of implant longevity and the need to address the causes of long-term failures [2].

As a possible cause of long-term THA and TKA failure, mechanical loosening due to loss of osseointegration or loss of periprosthetic bone stock is mainly linked to the stress shielding phenomenon and to wear-related osteolytic reactions [3,4].

Different strategies have been adopted to mitigate these effects, for example, reducing the structural stiffness of the prosthesis, using materials with a lower elastic modulus, and introducing porosity in part of the prosthesis [5,6]. Porosity can both lower the stiffness of the prosthetic implant and improve its primary and secondary fixation [7].

In recent decades, the orthopaedic community has witnessed huge improvements in operative techniques, digital preoperative planning, robotic-guided surgery systems, augmented reality, artificial intelligence algorithms, patient-specific instrumentation, and computer-assisted navigation. However, the evolution of prosthetic materials in joint replacement was slow until the introduction of additive manufacturing (AM) [8,9]. AM has revolutionized the way orthopaedic implants are designed and manufactured by allowing for the creation of better optimized highly porous structures that improve primary fixation, promote bone ingrowth, and enhance bone–implant biomechanics [10]. Since the introduction of 3D-printed highly porous titanium in the orthopaedic industry as a new benchmark for implantable devices, the clinical applications of this technology have been rapidly expanding to different devices in the AM era [11,12].

This perspective article describes some of the latest advancements in the clinical application of 3D printing SLM in the orthopaedic field. We focus on hip and knee arthroplasty, giving insights into the possible biomechanical advantages of these innovative solutions over the current state of the art and discussing future outlooks regarding possible novel prosthetic designs.

## 2. Additive Manufacturing with Powder Bed Fusion

AM, commonly known as 3D printing, creates three-dimensional parts by adding material layer by layer from digital models, in contrast to traditional subtractive (material-removal) or formative (casting, forging, moulding) techniques. This approach makes it possible to build complex shapes—that would otherwise be impossible to construct using conventional methods—and to merge multiple components into a single part, minimizing material waste and cut tool costs [11,12].

Nowadays, in the field of orthopaedics, implantable prosthetic components are also manufactured using AM technologies, particularly powder bed fusion (PBF), which features a heat source that selectively melts or sinters areas of a powder bed. Major PBF technologies include electron beam melting (EBM), selective laser melting (SLM), and selective laser sintering (SLS). Each of these PBF technologies has its own process parameters and characteristics which lead to certain advantages and limitations (Table 1).

While SLS is commonly used for rapid prototyping of plastic models for preoperative surgical planning or for patient-specific instrumentations, EBM and SLM are used to 3D print implantable metal prostheses for joint replacement [12]. In particular, with SLM technology, it is possible to produce metallic components under a controlled atmosphere directly from a 3D CAD file. The digital model is “sliced” into ultra-thin cross-sectional layers with a thickness corresponding to the powder bed thickness (20–100 µm). Each layer is selectively melted by a high-energy laser that completely fuses the metal particles of specific areas of the powder bed according to the cross-sectional design of the defined component geometry. The component is built in its entirety by melting each cross-section layer by layer. Loose, unfused powder remains in place after the PBF process is completed. This excess powder is then removed, reprocessed, and reused for a limited number of cycles.

## 3. Three-Dimensionally Printed Highly Porous Structures

One of the greatest strengths of AM, particularly for PBF when applied in orthopaedics, is the ability to create biomimetic 3D highly porous metal structures which closely resemble the trabecular architecture of cancellous bone.

Open, fully interconnected porosities of 50–70% and mean pore sizes within the range of 300–600 µm are considered ideal for optimizing bone ingrowth into the implant porosity [13,14,15]. Irregular porous structures created in a random way can favor permeability and therefore allow for bone ingrowth when compared to porous structures with regular cells [16,17]. In vivo osseointegration research and clinical research on retrievals have shown how 3D-printed highly porous titanium implants are able to promote fast bone ingrowth and osseointegration in cancellous and cortical bone because their structure and biomechanical properties resemble those of trabecular bone [18,19,20,21].

The dimension of the pores is strictly related to the dimension and geometry of the trabeculae forming the porous structure. Such pore dimensions, in the order of hundreds of microns (400–1000 μm) [22,23], represent a challenge when considering the printable dimension limits imposed by PBF technologies. The minimum dimension printable with Titanium–6Aluminum–4Vanadium (Ti6Al4V) powder is approximately 100–200 μm for SLM and the EBM technologies. With SLM, it is possible to print highly porous structures with a higher melting resolution, precision, and accuracy, leading to a higher porosity, more complex geometries, and lower trabeculae dimensions in comparison to EBM. On the other hand, SLM has lower processing speeds, higher residual internal stresses and a higher energy consumption than EBM.

## 4. Rationale for Novel Off-the-Shelf Cementless Prosthetic Components

These PBF technologies could be further leveraged for various novel clinical applications in order to include highly porous titanium structures in hip and knee implants that are commonly manufactured using traditional materials and technologies. The requirements for conceiving and developing such novel implants are to avoid the use of bone cement, Cobalt–Chromium–Molybdenum (CoCrMo), and stainless-steel (SS) alloys in order to improve primary and secondary stability through biological fixation, reduce stress shielding, avoid increasing wear, and avoid releasing reactive debris responsible for bone resorption reactions.

Cemented fixation with polymethylmethacrylate (PMMA) still represents the gold standard for anchoring certain prosthetic components to the bone, and is used in total knee arthroplasty and total shoulder arthroplasty. However, cemented fixation may jeopardize the longevity of the implant itself in some circumstances where the design of the cemented component or the bone anatomy do not allow for good mechanical anchorage. Bone cement is a polymer that acts like a filler, not as a chemical bonding agent. Anchorage between the cement and the implant, as well as between the cement and the bone, occurs through mechanical fixation. Thus, when the fixation of a cemented prosthesis is not favored by the anatomy of the bone segments, such as in long bones, the prosthetic components need to increase the area of the implant–cement interface through slots or grooves that improve mechanical locking [24]. With an improved cementing technique, cemented implants can also withstand lower shear stresses in comparison to osseointegrated highly porous implants, as easily observed in in vitro pull-out tests [25,26]. This aspect is of particular interest when the direction of loading is not perfectly perpendicular to the bone implant interface.

Poor cementing techniques have been associated with serious intraoperative complications due to fat and gas embolism, known as bone cement syndrome [27]. Temperatures above 47 °C for more than one minute during the exothermic PMMA polymerization reaction can cause bone tissue necrosis, compromising the integration of the implant with the bone [28,29]. Moreover, bone cement fragments that are not completely removed after cementation can be released into the joint capsule, leading to an increase in wear due to the third body mechanism [30]. Cemented prostheses can also be difficult to remove in case of revision [31].

Regarding implant raw materials, CoCrMo and SS are well-established biomaterials and have been largely used since the 1950s. CoCrMo is considered as the material of choice for manufacturing articulating components due to its mechanical and tribological properties. However, the elastic moduli of CoCrMo and SS are twice the elastic modulus of titanium alloy, meaning that CoCrMo and SS carry much more load than the surrounding bone, which is shielded by the metal component. Therefore, the periprosthetic bone is under reduced stress than it is in physiological conditions [32,33]. This phenomenon, known as stress shielding, leads to periprosthetic bone resorption, with a lower bone mineral density. A lower mineral density has been noted following TKA behind cemented CoCrMo femoral components a few years after surgery [34,35].

CoCrMo and SS components usually have cemented fixation, especially in TKA. Cementless fixation of these prostheses is achievable via coating with plasma-sprayed titanium particles and hydroxyapatite (HA) or by sintering titanium beads or mesh, other types of porous coatings, or highly porous tantalum structures. However, the primary stability of cementless CoCrMo or SS implants may be sub-optimal, mainly in elderly patients, with osteoporotic bone tissues [36].

CoCrMo and SS can release metal ions and byproducts into the surrounding tissues due to corrosion and other corrosion-assisted mechanisms such as tribocorrosion, fretting corrosion, crevice corrosion, and galvanic corrosion. The metallic elements released, typically Nickel, Cobalt, Chromium, and Molybdenum, can potentially elicit rare but serious adverse biological cell-mediated reactions, which in most cases lead to implant failure and revision [37].

Therefore, uncemented highly porous titanium implants could reduce stress shielding by promoting greater load transmission to the bone, thereby encouraging more physiological bone remodeling. They could also improve anchorage to the bone through both enhanced primary stability and secondary osseointegration. The trabeculae of 3D-printed highly porous structures increase friction with the cancellous bone, thus improving the initial press-fit anchorage. This immediate primary stability is essential to allow secondary osseointegration without deleterious micromotions, which could lead instead to fibrous tissue formation at the bone–implant interface. Highly porous prosthetic components have been particularly indicated for young active patients, as well as in elective procedures for elderly patients with osteoporotic bone or for complex cases where the implant’s ability to achieve a reliable bone fixation is essential [38,39].

However, a major limitation of 3D-printed highly porous titanium implants is that titanium alloys are not suitable to be used together with ultra-high-molecular-weight polyethylene (UHMWPE, or, simply, polyethylene) due to their poor tribological properties.

## 5. Articular Coupling of Ceramic-Coated Titanium Implants with Polyethylene

If a Ti6Al4V component is intended to be coupled with an UHMWPE component, its articulating surface must be treated to improve its poor tribological properties. Although it is highly biocompatible, Ti6Al4V’s surface is covered by a thin layer of titanium oxide (TiO_2_), which is responsible for an increase in polyethylene wear. Typical mechanisms of polyethylene wear against a titanium surface are adhesive wear and third body wear due to the titanium oxide debris produced by the articulating counterpart under loading conditions. A new oxide film is immediately formed over the worn areas of the titanium surface, and then a cyclic mechanism takes place, accelerating polyethylene wear [40]. To overcome this limitation, Ti6Al4V articulating surfaces must be treated. One possible treatment is to apply thin nitride-based coatings such as Titanium nitride (TiN), Titanium–Niobium nitride (TiNbN), and Zirconium nitride (ZrN). These so-called ceramic coatings provide a high surface hardness, chemical stability, and acceptable tribology for use with UHMWPE [41]. Thanks to these surface treatments, additively manufactured titanium components can be used as hard bearings together with polyethylene bearings, combining the benefits of both osteoinductive and osteoconductive highly porous titanium structures at the bone interface, with comparable wear performances to traditional articulating hard materials such as CoCrMo, SS, or ceramics against polyethylene.

The current clinical evidence for TKA shows that these ceramic-coated implants do not exhibit improvements over standard uncoated implants, but do exhibit comparable survival rates and clinical and radiographic results [42,43,44].

In a recent pin-on-disc study, TiNbN-coated titanium was able to reduce the wear rate of different types of UHMWPE in comparison to TiN-coated titanium [45]. A possible reason could be the lower friction coefficient of TiNbN in lubricated conditions due to the higher absorption of albumin in comparison to TiN [46]. However, ceramic-coated components do not improve polyethylene wear in comparison to uncoated components. A slight increase in polyethylene wear can occur when used in conjunction with a ceramic-coated component, as observed in in vitro knee wear testing with TiNbN-coated CoCrMo femurs compared to uncoated femurs [47].

Nonetheless, the polyethylene wear rate is not the only factor to consider when evaluating the tribology of joint replacement and its biological effects [48]. The osteolytic reaction is influenced not only by the overall volume of polyethylene wear per year, but also by the number, size, and oxidation degree of the polyethylene wear debris, in addition to other mechanisms of biomaterial wear such as metal ion release [48,49].

The osteolytic biological response has been proven in vitro to be significantly reduced in the presence of wear debris from vitamin E-stabilized UHMWPE compared to debris from non-stabilized polyethylene [50,51].

## 6. Novel Clinical Applications of 3D-Printed Highly Porous Titanium

Observations of materials, fixation methods, and biocompatibility show that there is sufficient reason to shift the attention towards further clinical applications of 3D-printed titanium alloy components with highly porous structures, besides their current clinical 3D printing applications in the orthopaedics field. Joint replacement should be a particular focus.

Based on these insights, a highly porous trabecular lattice was developed and produced by SLM in 2016 (Traser^®^, Permedica Orthopaedics, Merate, Italy). Unlike regular geometric lattices, Traser’s structure is random and based on the Voronoi tessellation algorithm [52,53]. Voronoi tessellation is a 3D space partition useful for modeling and describing various natural patterns, and its use has been proposed to model cancellous-bone-like three-dimensional porous structures [54]. This random lattice generates irregular-shaped pores (mean size: 520 µm, range: 100–2000 µm) and trabeculae (mean diameter: 250 μm), which form tortuous 100% interconnected channels. The 70% porosity favors osteoblast colonization and optimizes new bone growth inside the trabecular structure [25]. Despite its random design, the Traser lattice is macroscopically uniform, leading to homogeneous isotropic mechanical properties, with a Young’s modulus and Poisson’s ratio (1.5 GPa and 0.35, respectively) comparable to those of cancellous bone (Figure 1).

The metal powder employed for powder bed fusion (ASTM F2924-14) is medical-grade titanium alloy (Ti6Al4V), chosen for its optimal balance of biocompatibility, osseointegration capacity, lightness, mechanical properties and low allergenicity. The SLM process is carried out using high-energy Ytterbium fiber lasers (400–500 W) under a controlled Argon atmosphere. The process is managed in-house with dedicated machines and validated procedures, which guarantee a high quality and reproducibility of the final product.

## 7. Post-Processing Treatments

SLM represents only the first step towards the final product. After being printed (green state), titanium components require additional manufacturing operations to optimize their mechanical properties and surface finish and to obtain the final prosthetic design. The typical post-SLM workflow comprises several key stages.

Thermal treatment (annealing): Immediately after 3D printing, the parts are heated at 800 °C for at least 4 h and then slowly cooled over 24 h to room temperature. This treatment relieves residual stresses in the sintered material, increasing the alloy’s fatigue strength and toughness.Electrical discharge cutting: The printed components are separated from their building platform using a precision electrical discharge wire, which detaches the printed prostheses without introducing significant stresses or thermal alterations.Grit-blasting finishing: The titanium lattice contains incompletely melted titanium particles on its surface. Controlled grit-blasting with microsphere media (based on ZrO_2_, glass and Al_2_O_3_) removes superficial residual particles without damaging the porous structure. Besides cleaning the lattice, this process increases the trabecular surface microroughness to Ra 3 µm, which is known to stimulate osteoblast adhesion and differentiation, promoting rapid and robust osseointegration (Figure 2).Precision machining: Certain geometries or functional interfaces of the prosthetic device are refined by conventional subtractive techniques (milling, turning, grinding).Articular surface coating: The articular surfaces of Ti6Al4V bearing components are coated with a thin (4 mm ± 1.5) monolayer titanium–niobium nitride (TiNbN) coating via cathodic arc physical vapour deposition (PVD).Surface finishing: The metal articulating surfaces of the bearing components are mirror polished. This operation ensures that the articular portions meet the surface roughness requirements. After the PVD process, the coated surfaces undergo further surface finishing to remove residual nanodroplets left over from the coating process, thus reducing the surface roughness to Ra 0.03 mm (Figure 3).

Since 2016, Permedica Orthopaedics has used SLM to produce 3D-printed off-the-shelf and customized highly porous titanium hip, knee and shoulder components intended for cementless fixation. Recently, some novel off-the-shelf implants have been developed combining SLM highly porous titanium, TiNbN coatings, and vitamin E-blended UHMWPE with the purpose of finding biocompatible, long-lasting implants. Following this, Permedica Orthopaedics introduced a novel monobloc dual mobility cup and a novel femoral component onto the market in 2021, both 3D-printed by SLM using titanium alloy, coated with TiNbN, and mixed with a vitamin E-blended UHMWPE (Table 2).

## 8. Highly Porous Titanium Monobloc Dual Mobility Cups

Standard dual mobility cups (DMCs) are traditionally manufactured with CoCrMo and SS. The latest generation of such implants features both cemented and cementless press-fit cups with plasma-sprayed titanium, HA or Calcium Phosphate Apatite (CPA) coatings.

Recently, highly porous DMCs have been manufactured through constructing a standard 3D-printed highly porous titanium acetabular shell with a modular metal liner, typically made from CoCrMo, to allow for their use with a standard mobile PE head for dual mobility. These modular DMC constructs, although offering the advantage of a highly porous surface for enhanced primary and secondary stability, have been shown to be far from standard monobloc DMCs [55,56].

Preliminary clinical studies confirmed the osseointegration ability of a novel highly porous titanium DMC used in elective and trauma THA (Figure 4) [57,58].

This novel DMC, being additively manufactured via SLM using Ti6Al4V powder, has the unique feature of a highly porous titanium surface (Traser) integrated on the bone side, which improves primary cup fixation thanks to the surface high friction (Figure 5). The shape of this novel cup is similar to previous standard monobloc DMCs, but with an important polar flattening featuring a dual-radius cup profile to improve the initial equatorial press-fit, a symmetrical design to ease surgical implantation, and a reduced height between the cup opening plane and polar apex for medial bone sparing.

The cup articular side is PVD coated with TiNbN to allow for use with a standard vitamin E-blended polyethylene mobile liner. Due to the low contact pressure (due to bearing conformity) and, most importantly, the reduced working time of the large articulation in comparison to the small articulation, no coating delamination, breakthrough, or damage is expected [59,60]. Moreover, during function of the large articulation is working in activities involving wide hip movements, the mobile polyethylene head rotates inside the acetabular shell, pushed by the femoral neck and thus no high loading peaks are expected in the hip. On the contrary, high loads are involved during small articulation, i.e., during daily activities such as walking, running, and going up and down the stairs [61].

## 9. Highly Porous Titanium Femurs for Total Knee Arthroplasty

Cementless TKA has recently been gaining more support within the orthopaedic community due to new 3D printing technologies, which allow for the production of improved tibial components for biological fixation [62]. Cementless TKA with modern prosthetic designs leads to better survival rates even in high-demand patients such as active young males or morbidly obese patients [62,63]. Up to now, 3D printing has been used to manufacture cementless highly porous titanium tibial baseplates, with excellent clinical results. However, cementless femoral components are still manufactured by cast CoCrMo with porous titanium, trabecular tantalum, or HA coatings (Table 2) [64,65]. CoCrMo is known to be linked to the stress shielding phenomenon, which progressively reduces the mineral bone density in the femoral metaphysis and leads to metal hypersensitivity reactions in rare cases [35].

The concept of 3D-printed highly porous titanium cementless femoral components was born from the desire to remove CoCrMo from the knee. Modeling studies have confirmed that 3D-printed highly porous implants not only facilitate bone ingrowth, but also lower overall implant stiffness, mitigating the adverse bone remodeling caused by stress shielding compared to solid metal prostheses [7]. The mechanical and physical properties, modulus of elasticity, Poisson’s ratio, and compressive force resistance of the Traser lattice are very close to those of cancellous bone and, therefore, a titanium implant with a highly porous portion at the bone interface should reduce the stress shielding [66].

The femur design was derived from a cruciate-retaining CoCrMo femur design, maintaining the same shape, cylindrical pegs, and J-curve sagittal profile. The component is fully 3D printed with Ti6Al4V via SLM, featuring a Traser structure on the trochlear shield and condyles bone side with no continuity with the solid bulky portion. The highly porous portion is fully included within the component, leading to minimal interference with the bone. The high friction of trabecular spikes guarantees a strong primary stability.

The condyles articular surface is PVD coated with TiNbN to allow it to be used with a vitamin E-blended non-crosslinked polyethylene-fixed cruciate retaining tibial insert. In cruciate-retaining fixed-bearing TKA, where the similarity between articular surfaces is low, bearing wear is mainly influenced by contact pressure. Thus, in this clinical application, concerns about coating wear and damage are justified [67]. The coating adhesion is considered of utmost importance, as its delamination could cause excessive wear through third body wear. In addition, coating integrity (with no complete breakthrough) is mandatory so that the titanium substrate does not become exposed, which could increase polyethylene wear. No delamination or breakthrough of the TiNbN coating was observed over the condyles of this novel implant after 5 million cycles of a knee simulator wear test [68].

This novel device has started to be clinically investigated, with promising preliminary results (Figure 6) [69].

## 10. Future Perspectives

Due to the preliminary results from preclinical and clinical research and the almost complete lack of complaints from the field after four years of using these novel 3D-printed, highly porous titanium bearing components in clinical practice, extending the use of SLM to other possible clinical joint replacement applications has been proposed, following the same concept of replacing the CoCrMo or SS metals with highly porous titanium. Currently, a novel 3D-printed partial knee prosthesis intended for cementless unicompartmental knee arthroplasty (UKA) is under development.

The rationale for novel, highly porous all-titanium partial knee implants is the good results after the introduction of cementless UKA [70]. Traditional cementless UKA, in which a porous coating of titanium and hydroxyapatite is added to both the femoral and tibial components, has been shown to reduce the prevalence of radiolucencies compared to cemented UKA. Its other advantages are the reduced surgical time and the lack of cementation complications [71]. National registry data demonstrate comparable functional outcomes, improved survival rates, less aseptic loosening, and lower revision rates with cementless UKA [72]. Three-dimensional printing is not a novel manufacturing technique for partial knee prostheses. A recent study reported the long-term good survival and clinical results of a 3D-printed UKA which comprises a 3D-printed CoCrMo femoral component and titanium alloy tibial tray, with the ingrowth surface featuring an average interconnected porosity of 700 μm with a 40 μm hydroxyapatite coating [73].

The purpose of a novel 3D-printed titanium partial knee would be an optimized anatomical design for femoral and tibial components, with pegs for primary fixation and a highly porous titanium lattice on the bone side for secondary osseointegration and better load transfer to the tibial plateau [74]. The femoral and tibial pegs should be highly porous in order to allow for greater friction against cancellous bone, thus providing good primary stability (Figure 7). This would mainly be for the tibial plateau, which is the component most at risk of loosening in UKA [75].

As for the bicompartmental femur prosthesis, this partial knee 3D-printed titanium femur should cementless be TiNbN coated for compatibility with a fixed vitamin E polyethylene liner.

Besides UKA, other potential fields for novel clinical applications of SLM in knee or hip replacement could be investigated, such as cementless total hip resurfacing or patella-trochlear implants. In fact, these kinds of implants are traditionally based on cemented CoCrMo bearings. Therefore, using the same SLM method to obtain highly porous titanium components, as well as ceramic coatings to treat titanium articular surfaces and enhanced polyethylene for articulation, it could be possible to design highly porous titanium ceramic-coated femoral components for total hip resurfacing or highly porous titanium ceramic-coated trochlear shields for patellofemoral arthroplasty.

## 11. Conclusions

Novel off-the-shelf, 3D-printed, highly porous titanium implants appear to offer an innovative and promising solution in primary elective procedures, and are particularly indicated for young active patients and elderly patients with osteoporotic bones in acute procedures, as well as for complex cases where the implant’s ability to achieve reliable bone fixation is essential. The preliminary findings regarding the use of these novel implants in hip and knee arthroplasty are encouraging.

In future studies, the rationale and principles of these novel implants should be confirmed via short-to-mid-term clinical and radiographic results. In particular, clinical studies should investigate their longevity, safety, supposed superiority, and clinical relevance for patients in comparison to the current state of the art in total hip and knee arthroplasty.

## Figures and Tables

**Figure 1 biomimetics-10-00634-f001:**
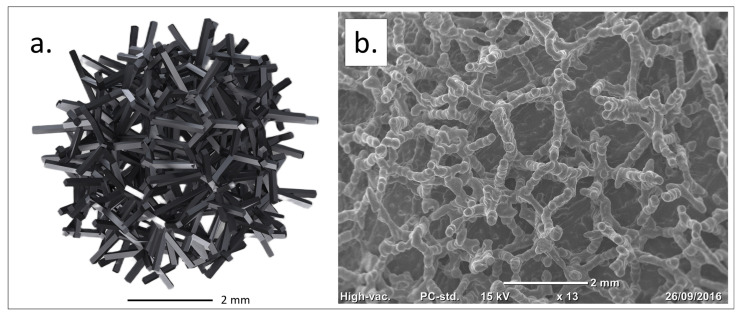
Traser structure. Courtesy of Permedica Orthopaedics. (**a**) Modeling of the randomized trabecular lattice created using the Voronoi-based space partitioning method. (**b**) Scanning electron microscopy (SEM) image of the randomized trabecular architecture.

**Figure 2 biomimetics-10-00634-f002:**
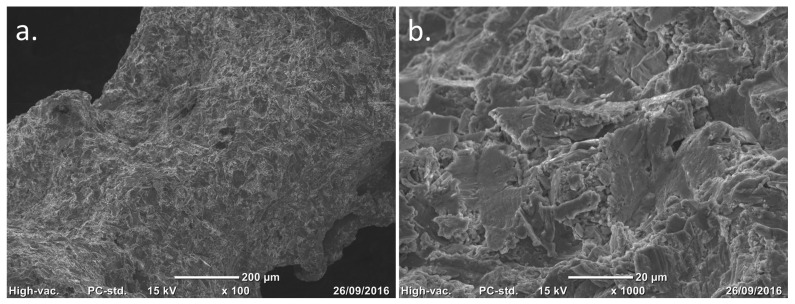
Scanning electron microscopy (SEM) images of the trabecular titanium surface. Courtesy of Permedica Orthopaedics. (**a**) Surface topography of trabeculae. (**b**) ×10 magnification of the same image showing details of the sandblasted microroughness.

**Figure 3 biomimetics-10-00634-f003:**
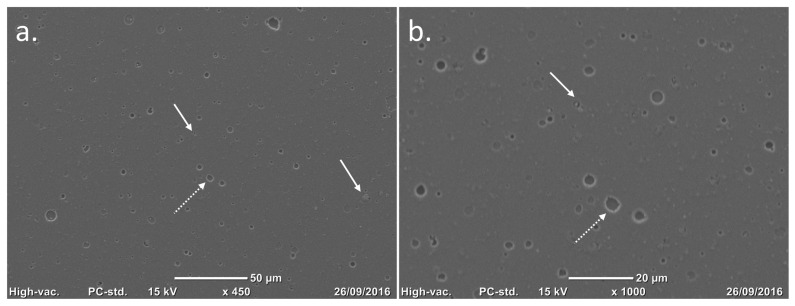
Scanning electron microscopy (SEM) images of a TiNbN coating surface. Courtesy of Permedica Orthopaedics. (**a**,**b**) After the mirror polish finishing, only nanodroplets (white arrows) and nanopores (white dotted arrows) are left on the titanium surface.

**Figure 4 biomimetics-10-00634-f004:**
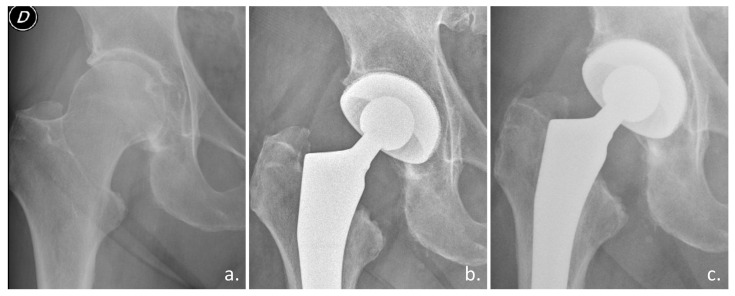
Clinical case with the Acorn Traser dual mobility cup. Courtesy of the Dept. of Orthopaedic Surgery of Ospedale Maggiore Carlo Alberto Pizzardi. A 76-year-old man with (**a**) right hip osteoarthritis, a 26 kg/cm^2^ BMI, and a history of diabetes, hypertension, benign prostatic hyperplasia (D indicates the right side). (**b**) One-year after the procedure, the cup appears well osseointegrated. (**c**) After three years, the patient could walk unaided and had experienced no episodes of subluxation.

**Figure 5 biomimetics-10-00634-f005:**
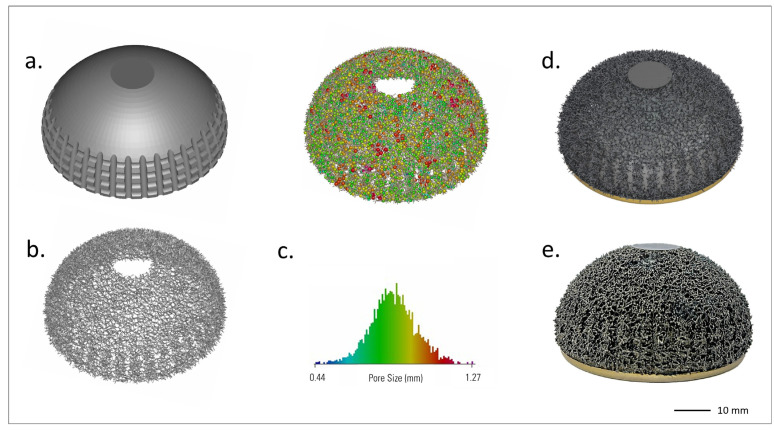
Steps in the selective laser melting (SLM) process for the Acorn Traser dual mobility cup. Courtesy of Permedica Orthopaedics. The (**a**) bulk portion of the component is input into the SLM machine, then the proprietary algorithm builds the (**b**) randomized trabecular portion on top. (**c**) The trabecular porosity of the model can be assessed. Each pore is approximated with the most appropriate sphere. These spheres have a range of diameters, color mapped as shown in the histogram. It should be noted that the pores are irregular and thus their maximum width might not match the sphere diameter. (**d**) Product rendering and (**e**) image of the final product.

**Figure 6 biomimetics-10-00634-f006:**
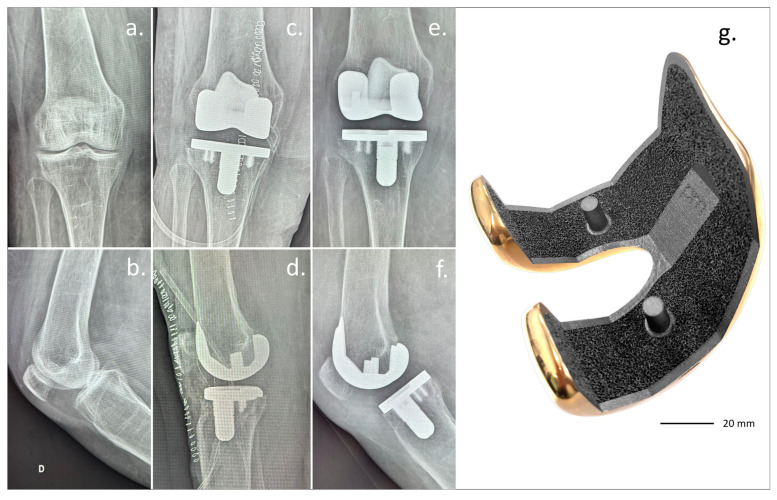
Clinical case with (**g**) the GKS Prime Flex Traser femoral component. Courtesy of Permedica Orthopaedics and the Dept. of Orthopaedic Surgery of Ospedale Maggiore Carlo Alberto Pizzardi. A 59-year-old woman with bilateral gonarthrosis, worse in the (**a**,**b**) right knee. (**c**,**d**) Post-operative and 6-month follow-up (**e**,**f**) radiographs showed complete osseointegration of the cementless implant, with no radiolucent lines or subsidence. Clinically, the patient reported excellent function and no pain in either knee.

**Figure 7 biomimetics-10-00634-f007:**
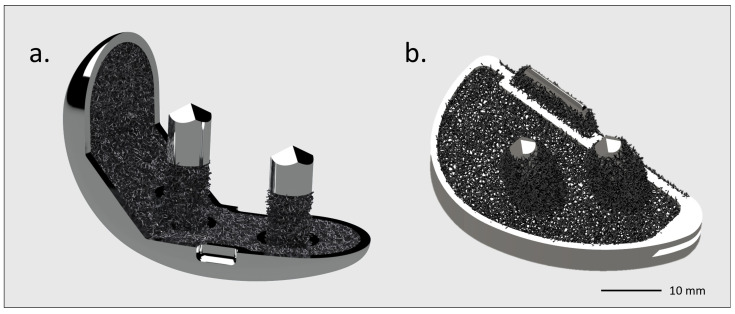
GKS One Evo Traser unicompartmental knee rendering. Courtesy of Permedica Orthopaedics. Cementless (**a**) femur and (**b**) tibia components with a highly porous structure, modeled on the bone side.

**Table 1 biomimetics-10-00634-t001:** Process parameters and main characteristics of the main powder bed fusion technologies.

Parameter/Characteristic	Electron Beam Melting	Selective Laser Melting	Selective Laser Sintering
Energy	Electron beam	Ytterbium fiber laser	CO_2_ laser
Atmosphere	Vacuum	Argon	Argon or nitrogen
Process temperature	700–1000 °C	30–170 °C	170–200 °C
Post-processing heat treatment	No	Yes	No
Powder bed material	Mainly metals	Mainly metals	Mainly polymers
Powder bed thickness	90–120 μm	20–100 μm	10–200 μm
Build platform	Yes	Yes	No
Melting spot	0.1–0.2 mm	0.08 mm	0.2 mm
Melting accuracy	±0.2–0.4 mm	±0.05 mm	±0.04 mm
Advantages	High process speed, less residual stresses	High precision and material density	No support needed, good mechanical resistance
Limits	Lower resolution than lasers, only conductive materials	Thermal stress, need for supports	Rougher surfaces, lower precision and accuracy than stereolithography

**Table 2 biomimetics-10-00634-t002:** Comparison between traditional implants and novel 3D-printed titanium implants for cementless monobloc dual mobility cups (DMCs) in total hip arthroplasty (THA) and for cementless femoral components in total knee arthroplasty (TKA). SLM, selective laser melting; TiNbN, titanium-–niobium nitride; CoCrMo, cobalt–chromium–molybdenum; SS, stainless steel; HA, hydroxyapatite.

	Monobloc DMC for THA		Femoral Component for TKA	
	New	Traditional	New	Traditional
Manufacturing process	SLM	Cast or forged	SLM	Cast or forged
Substrate material	Titanium alloy	CoCrMo or SS alloy	Titanium alloy	CoCrMo alloy, oxidized zirconium alloy, titanium alloy
Articular surface	TiNbN coating	Highly polished CoCrMo or SS, ceramic coatings	TiNbN coating	Highly polished CoCrMo, ceramic coatings, oxidized zirconium alloy, titanium nitride alloy
Bone side interface	Random highly porous titanium	Titanium/HA porous coating, Alumina	Random highly porous titanium	Titanium/HA porous coating, titanium sintered beads, highly porous tantalum coating *
Mean pore size	520 μm	100–250 μm	520 μm	100–250 μm (* 600 μm)
Porosity	70%, open, completely interconnected	20–50%, closed, not interconnected	70%, open, completely interconnected	20–50%, closed, not interconnected (* 80%, open, completely interconnected)

“*” explains that 600 microns mean pore size is referred to tantalum coating.

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
