# Peer review of "Novel Clinical Applications of 3D-Printed Highly Porous Titanium for Off-the-Shelf Cementless Joint Replacement Prostheses"

_biomimetics, 2025, doi:10.3390/biomimetics10090634_

Round 1
Reviewer 1 Report
Comments and Suggestions for Authors
This paper presented the novel clinical applications of 3D-printed highly-porous titanium ofr off-the-shelf cementless joint replacement prostheses. The newly introduced implants offer an innovative and promising solution, particularly indicated for young active patients, as well as elderly patients with osteoporotic bone, or for complex cases. The paper is well structured and well written, but the following issues should be addressed:
- In the title, the word “review” should be added to make it clear it is a review paper.
- For all the figures, please make sure the copyrights have been obtained.
- “2. Additive manufacturing and selective laser melting” is too few. More contents should be added.
- Some tables should be added to make the systematic comparison and summary.
- A conclusion part should be added in.
Reviewer 2 Report
Comments and Suggestions for Authors
This manuscript is well structured and informative, and presents developments in the field of 3D printed, highly porous titanium implants for cementless joint replacements. However, some points could be improved to strengthen the article.
Comments:
1- Please explain why SLM is particularly advantageous for this application. Please compare this technique with other AM techniques. In addition, the manuscript could benefit from a summary table or schematic that compares traditional implant designs with the newer SLM titanium implants.
2- Some important claims, such as those related to the wear resistance of TiNbN coatings or the osseointegration, would benefit from more references. Quite a few statements in the text are missing citations. For example, the statement in line 42 needs a reference. I recommend going through the manuscript again carefully to ensure that each scientific or clinical claim is properly supported. Here as well: claims about clinical outcomes (e.g., reduced loosening, better fixation, etc) would be stronger if backed by more clinical data and appropriate references.
3- In line 46, when mentioning improvements in operative techniques, etc, please consider providing specific examples and relevant references to support that.
4- Please provide references or clarify the data source for all the figures. In Figure 4a, please clarify what "D" is.
5- Although this is a "Perspective" article, the tone and content feel closer to a detailed technical review. It would be helpful to make a clearer distinction between scientific insights and specific product information.
6- In section ‘Future Perspective’, it would be great to include where the authors see this field in the coming years (e.g., opportunities, challenges, or if innovations are still needed)
7- Considering a section that acknowledges some of the current limitations (i.e., long-term clinical outcomes, durability, reproducibility in the manufacturing process, potential risks associated with TiNbN coating delamination, etc) is recommended.
8- Please avoid using statements with ‘expected’ or ‘favourable’ results. If possible, try to support these claims with clinical study results or literature.
9- Please include scale bars in the figures.
10- Please split or rephrase some of the sentences that are quite long.
Round 2
Reviewer 1 Report
Comments and Suggestions for Authors
All my issues have been addressed, thanks.
Author Response
We are grateful for the help you previously offered in improving our manuscript.
Reviewer 2 Report
Comments and Suggestions for Authors
I would like to thank the authors for addressing the concerns. It would be beneficial to include:
- Scale bars in Figures 1a, 5, 6g, and 7.
- A section titled "Conflict of Interests.
